# Non-Communicable Diseases Among Forcibly Displaced People: A Systematic Mapping Review

**DOI:** 10.3390/ijerph22010063

**Published:** 2025-01-05

**Authors:** Kyohei Nishino, Tshewang Gyeltshen, Mahbubur Rahman

**Affiliations:** Graduate School of Public Health, St. Luke’s International University, Tokyo 104-0044, Japan; 22mp207@slcn.ac.jp (T.G.); rahman@slcn.ac.jp (M.R.)

**Keywords:** non-communicable diseases, forcibly displaced people, systematic mapping review, diabetes mellitus, hypertension, public health, international cooperation, health policy

## Abstract

Background: Non-communicable diseases (NCDs) pose a serious global health challenge, accounting for 74% of all deaths worldwide, with low- and middle-income countries (LMICs) disproportionately affected. These challenges are further exacerbated in humanitarian settings, particularly among forcibly displaced people (FDP). Despite the critical need for NCD management in these populations, their epidemiology remains poorly understood. This highlights an urgent research priority to address knowledge gaps and improve their health outcomes. Methods: In this research, we conducted a systematic mapping review to aggregate and categorize existing publications on NCDs among FDP. Literature searches were performed across five electronic databases, namely PubMed, Cochrane Library, Embase, Global Index Medicus, and Google Scholar, using predefined criteria related to target populations, research domains, and study design. The evidence was systematically coded and analyzed to assess the current research status on NCDs among FDP. Results: A total of 310 publications were included in the review. The findings indicate an increasing trend in publications on NCDs among FDP since 2014. In contrast, most studies revealed low evidence levels. Disease-specific research primarily focused on diabetes mellitus (DM) (26.4%) and hypertension (19.8%), addressing health status (43.1%) and health policy (32.0%). Studies mainly concentrated on Syrian (45.5%) and Palestinian (18.9%) refugees, with limited research on other countries and types of FDP. Funding sources were mostly governmental (23.1%) and philanthropic foundations (17.6%), although many studies were unfunded (26.2%). Conclusions: This is the first systematic mapping review on NCDs among FDP. The findings revealed both current knowledge areas of focus and gaps. Although the research quantity has increased, the evidence quality remains low. The retrieved studies predominantly focused on DM and hypertension among Syrian and Palestinian refugees, causing significant knowledge gaps regarding other geographical areas and types of FDP. Future research should prioritize higher-quality studies, expand the geographical scope, and include diverse types of FDP.

## 1. Background

Non-communicable diseases (NCDs) present a serious global health challenge. According to the World Health Organization (WHO), NCDs are responsible for approximately 41 million deaths annually, accounting for 74% of all deaths globally [1]. Adding to the mortality, the morbidity associated with NCDs severely impacts the quality of life of millions due to long-term disability and exerts immense pressure on the healthcare system. These facts underscore an urgent need for comprehensive strategies to combat NCDs and necessitate robust, sustained, and coordinated international responses. At the helm of international efforts, the WHO developed the Global Action Plan and Control of NCDs 2013–2030 [2]. This strategic declaration is designed to address the prevention of and reduction in morbidity, mortality, and the myriad socioeconomic impacts related to NCDs, with an ambitious goal of a 25% relative reduction in premature mortality from NCDs by 2025. In addition, Sustainable Development Goal (SDG) 3, which focuses on ensuring healthy lives and promoting well-being for all ages, is particularly relevant to the health challenges faced by FDP [3]. It sets a clear target for reducing one-third of premature mortality from NCDs by 2030, obliging member states to reduce the risk factors for NCDs, such as tobacco consumption and substance abuse, and achieving universal health coverage (UHC).

Despite these international initiatives, progress faces significant challenges, particularly due to the serious disparities of NCD burdens across economic divides. High-income countries (HICs) have experienced a noteworthy decline in the disease burden since the 1980s, thanks to advanced healthcare systems and effective public health policies. However, low- and middle-income countries (LMICs) bear the brunt of the impact, with over 75% of global NCD-related deaths occurring in these regions [1]. This disparity is primarily attributed to a confluence of risk factors prevalent in LMICs, including unhealthy diets, high consumption of tobacco and alcohol, inadequate healthcare infrastructure, and limited political will to address the issues [2,4,5].

The challenges are further compounded in the settings of humanitarian crises, such as among forcibly displaced people (FDP), defined as people who are away from their home or home region “as a result of persecution, conflict, generalized violence or human rights violations” [6]. Recent statistics about FDP revealed a concerning trend: the number worldwide has surged dramatically over the last few decades, surpassing 100 million in 2022 [7]. Furthermore, the data reveal a disturbing reality where over three-quarters of these displaced individuals remain in protracted displacement situations, with the average duration extending beyond 20 years [8]. The long-term displacement presents serious challenges for managing NCDs, as healthcare systems are often overwhelmed by immediate demands such as infectious diseases and trauma. In these contexts, the chronic nature of NCDs is frequently neglected, and access to essential medications and ongoing care is severely limited. In addition to the direct health challenges, displaced populations face socioeconomic barriers that further complicate NCD management. The displacement often disrupts livelihoods, limits access to clean water and nutrition, and leads to overcrowded living conditions, all of which contribute to the worsening of chronic diseases. Moreover, the stress and trauma associated with displacement can exacerbate conditions such as hypertension and diabetes, creating a vicious cycle of worsening health outcomes. Despite the evident need for targeted interventions in these populations, previous reviews, including those by Blanchet et al. [9] and Ruby et al. [10], have highlighted a critical scarcity of research on NCDs in humanitarian contexts. Humanitarian responses have traditionally prioritized acute health needs, and chronic conditions remain underrepresented and poorly understood. Given the growing burden of NCDs and the limited academic focus in humanitarian settings, a systematic understanding of the current state of research is urgently needed. This research addresses these gaps by conducting a systematic mapping review of the literature on NCDs among FDP. Given the limited availability of high-quality empirical studies, this mapping review aims to synthesize and categorize existing evidence, providing a comprehensive overview of the current state of research. The primary goal is to aggregate and categorize available evidence, creating meta-data to draw a global picture of NCDs among FDP. By systematically mapping the existing literature, this research seeks to provide an extensive overview of the current state of knowledge and the scale of research conducted to date. This comprehensive aggregation helps facilitate a clearer understanding of how NCDs are addressed in humanitarian settings, particularly in FDP.

In addition, this research seeks to underscore knowledge focuses that can implement secondary synthesis and identify knowledge gaps that need further investigation. By providing current research focuses and gaps in this issue, the research holds significant potential to guide future studies, shape health policies, and ultimately contribute to better health outcomes for FDP suffering from NCDs. Consequently, the systematic mapping review serves not only as an academic exercise but also as a vital step toward improving the health and well-being of FDP.

## 2. Methods

### 2.1. Methodological Framework

There is no standard methodological framework for systematic mapping reviews in the public health field as it is still an evolving method of evidence synthesis. To ensure the research quality, this study applied a methodology in environmental sciences and software engineering that follows the “same rigorous, objective, and transparent processes as do systematic reviews” [11]. The systematic mapping review was structured into five distinctive stages: (1) defining the scope and research questions along with establishing inclusion and exclusion criteria, (2) conducting a comprehensive search for relevant evidence, (3) screening the gathered evidence for relevance, (4) coding the evidence based on the predefined definitions, and (5) synthesizing and presenting the findings.

### 2.2. Defining the Scope and Research Questions Along with Establishing Inclusion and Exclusion Criteria

This study proposes the six research questions (RQs) below to broaden and analyze data related to NCDs among FDP from multiple dimensions.

RQ1. Temporal trend: How much research on NCDs among FDP has been published each year?

RQ2: Scientific evidence level: How much research on NCDs among FDP has been published based on the scientific level of evidence?

RQ3. Research domains: Which diseases and domains of medical interventions are prioritized in research on NCDs among FDP?

RQ4. Geographical areas: Which host countries and countries of origin for FDP are the research focus on NCDs?

RQ5. Types of FDP: Which types of FDP and living environments are prioritized in research on NCDs among FDP?

RQ6. Funding source: Which entities have provided financial support for research on NCDs among FDP?

Detailed inclusion and exclusion criteria were defined to ensure a systematic and objective approach to selecting studies for this mapping review. These criteria were based on six aspects: the target population of the study, target diseases, the specific research domains relevant to NCDs among FDP, the adherence of the study design to criteria for the evidence level of “evidence-based nursing care guidelines” [12], publication types, and language. These criteria allowed for a structured and evidence-informed assessment of the literature. The detailed inclusion and exclusion criteria are described in Table 1. Table 2 shows the level of scientific evidence, which was defined based on “evidence-based nursing care guidelines” [12].

### 2.3. Conducting a Comprehensive Search for Relevant Evidence

An extensive search strategy was enacted to systematically gather studies on NCDs among FDP. We searched five prominent electronic databases from 22 November to 8 December 2022: PubMed, Cochrane Library, Embase, Global Index Medicus, and Google Scholar. The database selection was decided to gather the widest possible spectrum of research outputs, ranging from peer-reviewed articles to grey literature, ensuring the inclusivity of relevant studies.

Keywords were chosen for their relevance and the frequency of use in the existing literature and were used in conjunction with filters to refine the relevance and accuracy of the search results.

### 2.4. Screening the Gathered Evidence for Relevance

All the identified literature was imported into the Mendeley reference management software2.127.1 to facilitate systematic screening. Following the removal of duplicates, two authors independently screened the titles, abstracts, and full texts of the collected literature, applying the defined inclusion and exclusion criteria. Any discrepancies between the two authors were resolved through their discussion.

### 2.5. Coding the Evidence Based on the Predefined Definitions

The coding results were compiled in Excel and subsequently analyzed using Python 3. This systematic coding process was instrumental in structuring the evidence for in-depth analysis and synthesis, laying the groundwork for a comprehensive understanding of the current state of research on NCDs among FDP. Consequently, this process led to insightful interpretations and substantive conclusions from the collected research data.

### 2.6. Synthesizing and Presenting the Findings

The coding of the retrieved literature was conducted using nine key variables in line with the six research questions: year of publication, level of scientific evidence, target diseases, research domains, countries of origin, host countries, living environments, types of FDP, and funding sources. The percentages in figures were rounded to one decimal place for clarity.

## 3. Results

A total of 310 publications were extracted from 1891 that were identified through a defined screening process (Figure 1 and Figure 2) .Given the selected publications, the six research questions are discussed in this section. The retrieved 310 publications are shown in Appendix A.

RQ1. Temporal trend: How much research on NCDs among FDP has been published each year?

The first study about NCDs among FDP was published in 2009 (Figure 3). It was the WHO’s strategic plan for Palestinian refugees in the Gaza Strip. For several years, annual publications remained below ten; however, a growing trend has been observed since 2014. This trend became pronounced in 2017, with over 80% of the literature being published in the last five years.

RQ2: Scientific evidence level: How much research on NCDs among FDP has been published based on the scientific level of evidence defined by “evidence-based nursing care guidelines” [11]?

Approximately 70% of the retrieved literature fell into low evidence levels, VI or VII (Figure 4). Notably, there were no publications at level I, and only a limited number of studies were categorized as levels II and III. One publication was about retrospective cost analysis from provider perspectives.

Figure 5 illustrates the distribution of publications by evidence level over time. The results show both consistency and changes. Evidence level VI, with notable constancy, continued to be the most represented category throughout the evaluation period. However, remarkable shifts were observed in the relative proportions of evidence levels V and VII. Evidence level VII was the dominant category until 2016; however, it exhibited an apparent decline trend after 2016. In contrast, publications at evidence level V have been increasing, especially from 2020 onward, and it has become the second most frequent category.

RQ3. Research domains: Which diseases and domains of medical interventions are the research focus on NCDs among FDP?

The majority of the research described a comprehensive overview of NCDs (41.8%) (Figure 6). This result can be attributed to the high proportions of review articles (evidence level V) and expert opinions (evidence level VII) among the retrieved studies.

Among disease-specific research, DM (26.4%) and hypertension (19.8%) were the two most common focuses. The two diseases collectively comprised nearly half of the research targeting specific diseases. Subsequently, mental health (11.4%), CVD (8.4%), and cancer (6.6%) were studied; however, the number was limited. Others included oral health (4.0%), chronic respiratory diseases (1.5%), dyslipidemia (1.3%), musculoskeletal disorders (0.8%), and arthritis (0.8%).

As for the research domain, health status and health policy were predominant, comprising 43.1% and 32.0%, respectively (Figure 7). Compared to the two research focuses, less priority was given to patients’ experience (16.1%) and health system (8.8%).

RQ4. Geographical areas: Which host countries and countries of origin for FDP are the research focuses on NCDs?

Among the 222 publications defining the target countries of origin, almost half of the publications focused on Syria (45.5%), followed by Palestine (18.9%), Afghanistan (6.3%), and Myanmar (5.4%) (Figure 8). These four countries accounted for more than three-quarters of the publications. Meanwhile, publications about Venezuela and South Sudan, the major countries of origin of FDP, were only three and one, respectively.

The target host countries were more diverse than those of origin (Figure 9). However, among the top seven target countries, five (i.e., Lebanon, Jordan, Gaza, Syria, and West Bank) are under the UNRWA management. As Turkey is the main host country of Syrian refugees, the result is compatible with that of countries of origin. Interestingly, the USA ranked third among host countries, despite not being a primary host country for FDP.

RQ5. Types of FDP: Which types of FDP and living environments are the research focus on NCDs among FDP?

In the analysis of types of FDP, refugees were notably predominant, comprising 70.4% of the total (Figure 10). Other categories, namely asylum seekers, internally displaced people (IDP), and Venezuelans displaced abroad, accounted for much smaller proportions at 10.1%, 8.9%, and 0.9%, respectively. No study was found for stateless people.

The findings of living environments show a diverse distribution (Figure 11). The majority, comprising 54.5%, were categorized under “all”, indicating a general or unspecified living environment. This was followed by 29.7% residing in non-camp settings and 14.5% in camp-based environments.

RQ6. Funding source: Which organizations financially supported the research on NCDs among FDP?

The most common funding source was the government or government-related funding agencies, contributing to 23.1% of the total (Figure 12). This was closely followed by philanthropic foundations at 17.6% and academic institutions at 16.3%. International organizations, non-governmental organizations (NGOs) or non-profit organizations (NPOs), and industry and corporate research funding each constituted around 4% of the funding sources. Meanwhile, more than a quarter of the retrieved literature received no funding.

## 4. Discussion

To the best of our knowledge, this is the first systematic mapping review about NCDs among FDP. The findings revealed both our current knowledge focuses and the gaps in the concerning issue. They enable us to expand and deepen our discussion, providing a clearer view of the current situation, exploring the reasons behind the identified barriers, and developing tailored strategies to move toward more effective NCD control for better health outcomes.

### 4.1. Research Focuses

The temporal trends in the publications showed a clear upward trend, especially in the last six years. This indicates the growing recognition of the importance of addressing NCDs in this vulnerable population.

Research questions 3–6 have clarified the current research focuses within their respective topics. The predominant diseases studied are DM and hypertension, with research primarily concentrated on health status and health policy aspects. The main countries of origin for FDP studied are Syria and Palestine, whereas the host countries most frequently assessed are those under the UNRWA management, namely Lebanon, Jordan, Gaza, Syria, and the West Bank, alongside the USA and Turkey. The major type of FDP referred to in the retrieved studies is refugees. From these findings, it is evident that the most robust and detailed body of knowledge currently available pertains to DM and hypertension among refugees originating from Syria and Palestine, with a particular focus on their health status and health policy.

### 4.2. Research Gaps

Three major research gaps were identified through this review: quality of research, geographical coverage, target diseases, and type of FDP.

#### 4.2.1. Gap 1. Quality of Research

The overall quality of retrieved studies revealed a low level of evidence in research on NCDs among FDP. The majority of publications fell within evidence levels V (21.3%), VI (21.9%), and VII (46.8%). Despite an increase in the quantity of research, the temporal trend indicates that the evidence level remains consistently low. Collectively, evidence levels V, VI, and VII accounted for approximately 90% of studies, with a quite limited number of primary research studies over time.

#### 4.2.2. Gap 2. Geographical Coverage

Another concerning gap is the disproportionate emphasis on the countries of origin of FDP. While the majority of research focuses on Syria and Palestine, this overlooks the millions of FDP residing in other countries, notably LMICs, including Africa, Southeast Asia, South Asia, and South America [13] (Figure 13 and Figure 14).

Host countries were also significantly affected by the focus of research attention. As previously noted, five out of the seven frequently studied target countries were under UNRWA management. Conversely, among the top ten host countries of people displaced across borders, besides Turkey and those under UNRWA management, countries such as Colombia, Uganda, Pakistan, Germany, Sudan, and Bangladesh did not constitute the main focus of the retrieved studies [14] (Figure 15).

#### 4.2.3. Gap 3. Target Diseases

The findings revealed that the majority of research on NCDs among FDP was concentrated on DM and hypertension, with limited attention to other major NCDs such as cancers and chronic respiratory diseases. This contrasts sharply with the broader global NCD research, which emphasizes that CVDs, cancers, chronic respiratory diseases, and DM are the four main types of NCDs contributing to over 80% of all premature NCD deaths worldwide [1]. The disproportionate focus on DM and hypertension in FDP research suggests the underrepresentation of other critical NCDs that are likely prevalent within this population.

#### 4.2.4. Gap 4. Types of FDP

The findings further highlighted a notable research gap concerning the types of FDP. Over 70% of studies focused on refugees, who constitute only a quarter of the total FDP [14]. In contrast, IDP, who represent the predominant category of FDP, were the focus of less than 10% of the retrieved research (Figure 16).

These identified research gaps, ranging from quality of research, countries of host and origin, and target diseases to types of FDP, underscore critical oversights in the academic literature and policy analysis on FDP. These gaps suggest that certain diseases affecting FDP, as well as particular subgroups defined by their country of origin and host, have been neglected. Moreover, the scarcity of research on IDP indicates a substantial deficiency in our understanding of the complex challenges these individuals face. The absence of comprehensive high-quality research hampers the development of targeted interventions and policies, essentially rendering many FDP invisible within academic research and the broader international response to displacement issues. Consequently, their vulnerabilities continue unaddressed. Therefore, addressing these research gaps is imperative to foster a more inclusive and effective approach. Expanding the scope and depth of research to underrepresented diseases and populations will not only enhance the academic and policy framework but also improve the precision and impact of health interventions. Such efforts are essential to ensure that no groups remain marginalized in the global efforts to address displacement and its associated difficulties.

## 5. Limitations

This mapping review embodies several limitations that may impact the breadth and depth of the conclusions. Firstly, the exclusion of non-English articles represents a significant limitation. This language restriction can introduce a cultural and geographical bias, overlooking substantial research on NCDs among FDP conducted in non-English speaking countries. Secondly, the omission of unpublished data can lead to an underrepresentation of available evidence. Unpublished studies, including national health data and conference papers, often contain relevant data that are not available in the retrieved publications. Excluding unpublished data could overlook a comprehensive view of current practices and challenges, particularly in capturing innovative strategies that are in the experimental phase of implementation. Thirdly, publication bias causes another concern. The bias can lead to an overestimation of the effectiveness of interventions and health status among FDP, as studies with positive or significant findings are more likely to be published. Conversely, studies that do not demonstrate efficacy or highlight negative outcomes may be underreported, which is equally important for developing a full understanding of what does not work or requires improvement. Furthermore, FDP who lack access to healthcare in their host communities and countries are often underrepresented in the literature. Although publication bias was not formally assessed in this study, its potential impact was inferred based on the limited diversity and predominance of positive findings in the retrieved literature. This skewed dissemination of information based on overly optimistic data can mislead our understanding of the situation. These limitations suggest that, while this mapping review provides valuable insights into the issue of NCD management among FDP, the results should be interpreted with caution, especially the generalizability and depth of the findings.

## 6. Conclusions

This systematic mapping review synthesizes the current state of research on NCDs among FDP, revealing key insights into both the progress made and the gaps that remain. The growing academic interest in this issue, particularly since 2014, is encouraging, as it signals a recognition of the importance of addressing NCDs in vulnerable populations. However, significant gaps persist, both in the quality of the research and the populations and diseases studied. DM and hypertension dominate the research landscape, with less attention paid to other major NCDs such as cancer and COPD. Geographically, the research is heavily concentrated on FDP from Syria and Palestine, with populations from Africa, Southeast Asia, and South America largely overlooked. Furthermore, the majority of studies focus on refugees, while IDP, who comprise the majority of FDP, are underrepresented in the literature.

Addressing these gaps is crucial for advancing the global response to NCDs among FDP. Future research should prioritize higher-quality studies, expand the geographical scope of research, and include a more diverse forcibly displaced population. By doing so, policymakers and practitioners will be better equipped to develop evidence-based strategies that improve health outcomes for all FDP affected by NCDs. These efforts are vital to upholding the SDGs’ pledge of “no one left behind”, ensuring that FDP receive the necessary attention and care within humanitarian responses.

## Figures and Tables

**Figure 1 ijerph-22-00063-f001:**
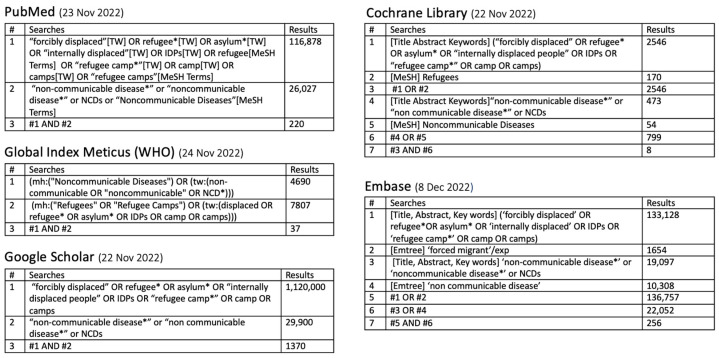
Result of database search. * is used as a wildcard character for database searches.

**Figure 2 ijerph-22-00063-f002:**
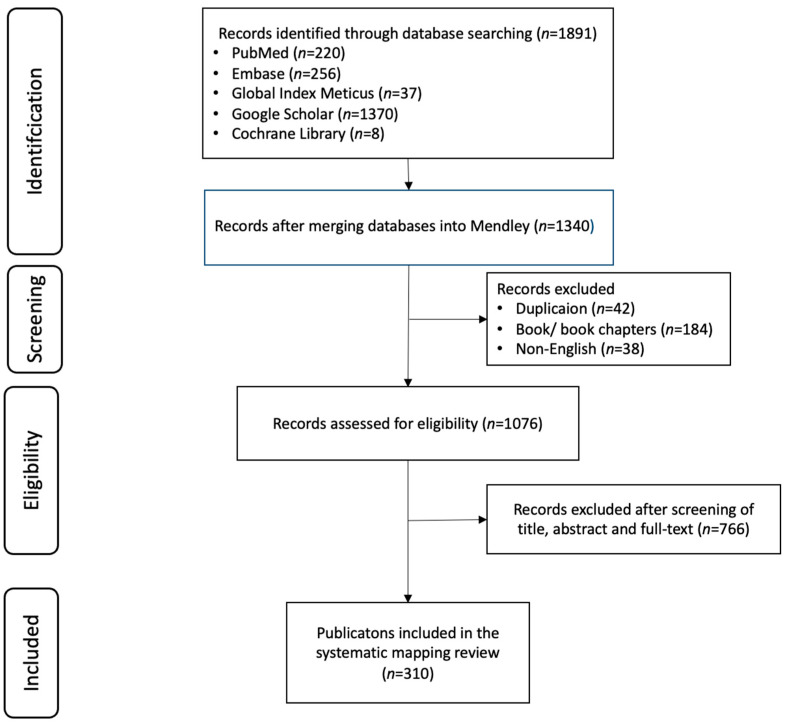
Flow diagram of publication selection.

**Figure 3 ijerph-22-00063-f003:**
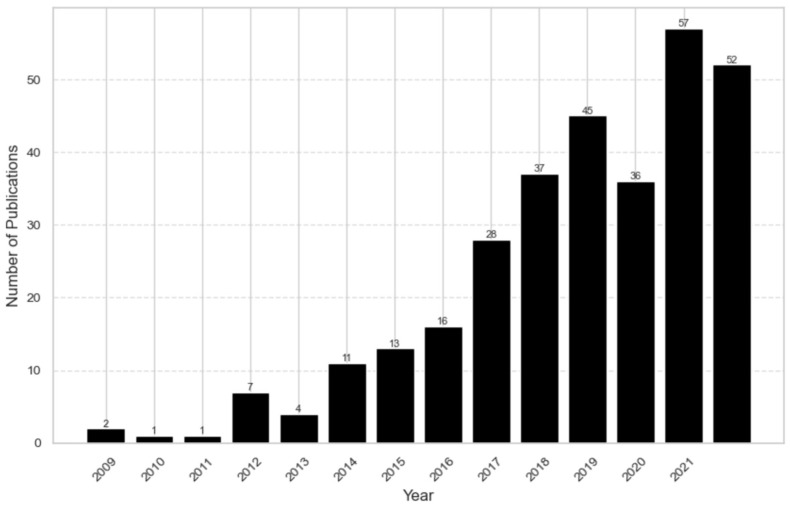
Temporal trend.

**Figure 4 ijerph-22-00063-f004:**
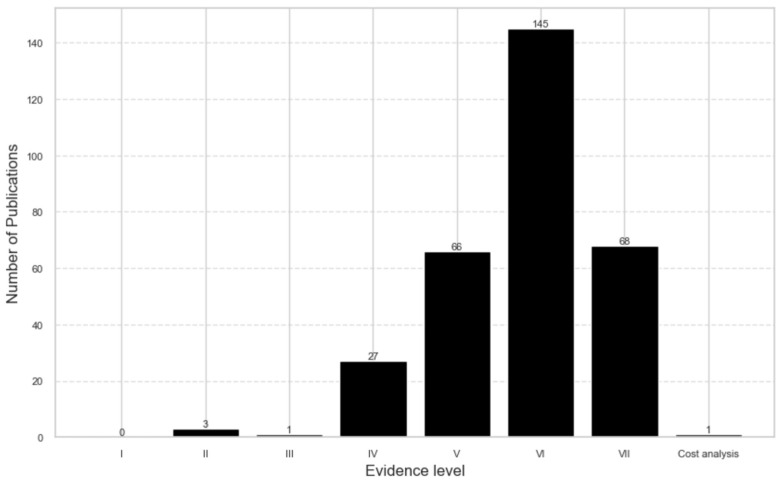
Scientific evidence level.

**Figure 5 ijerph-22-00063-f005:**
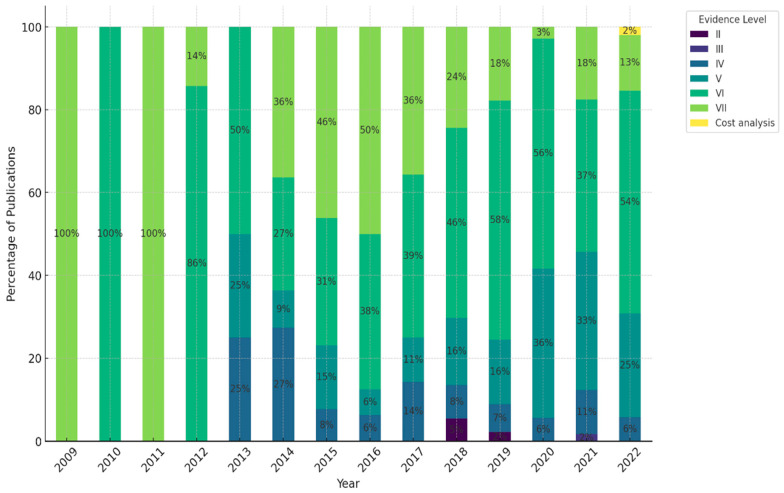
Scientific evidence level in each year.

**Figure 6 ijerph-22-00063-f006:**
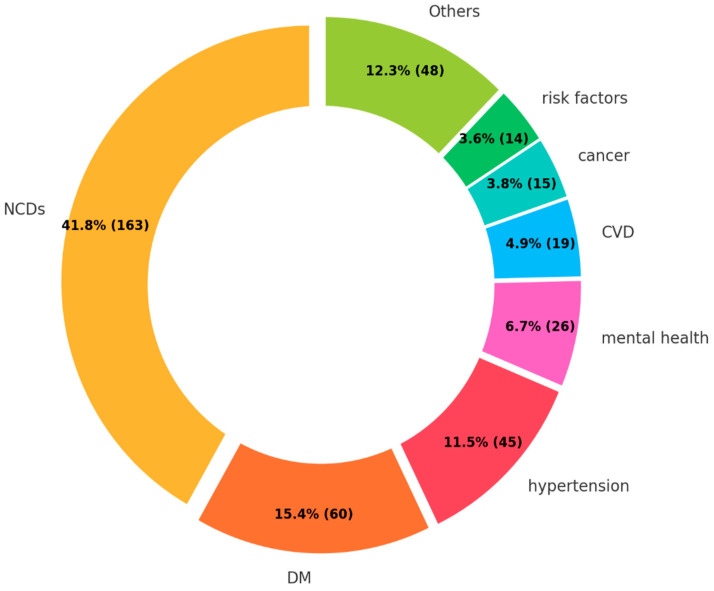
Target diseases.

**Figure 7 ijerph-22-00063-f007:**
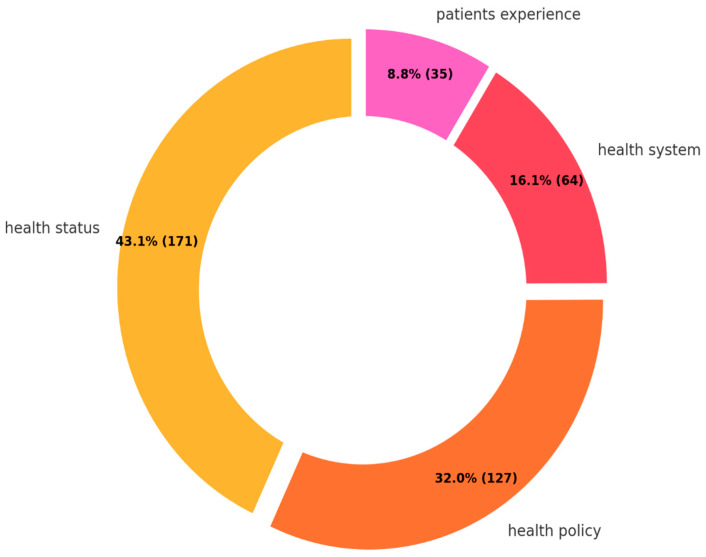
Research domain.

**Figure 8 ijerph-22-00063-f008:**
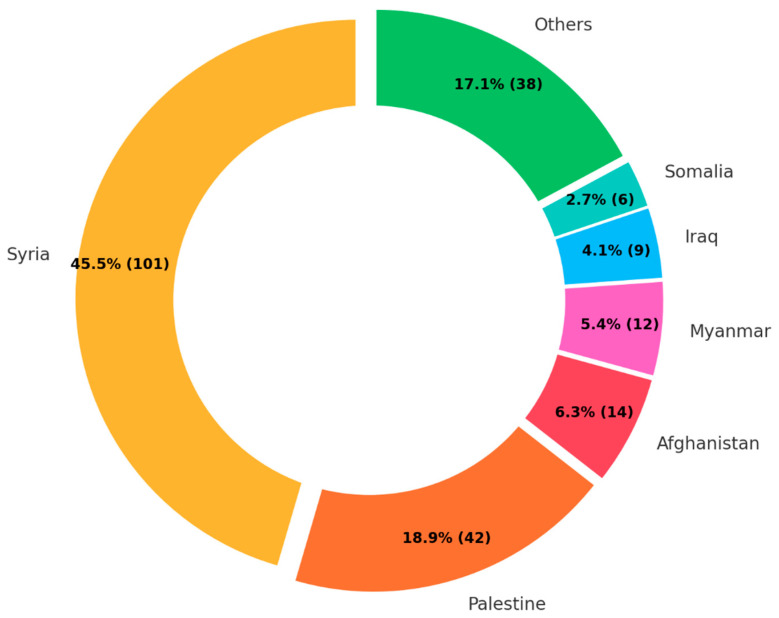
Number of publications by country of origin.

**Figure 9 ijerph-22-00063-f009:**
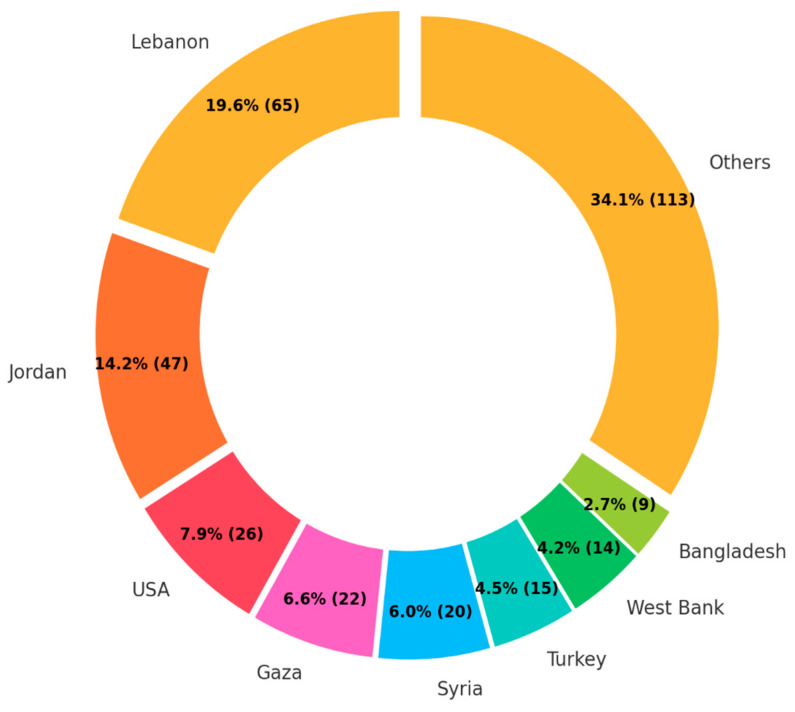
The number of publications by host country.

**Figure 10 ijerph-22-00063-f010:**
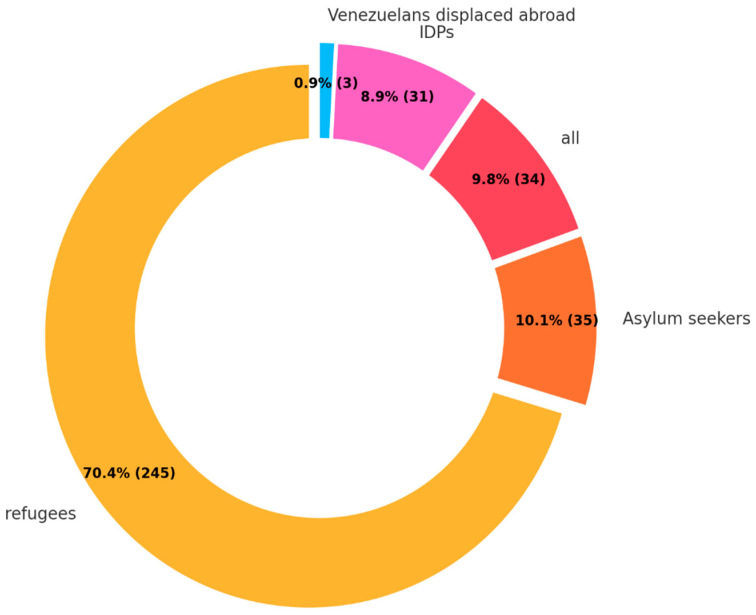
Type of FDP.

**Figure 11 ijerph-22-00063-f011:**
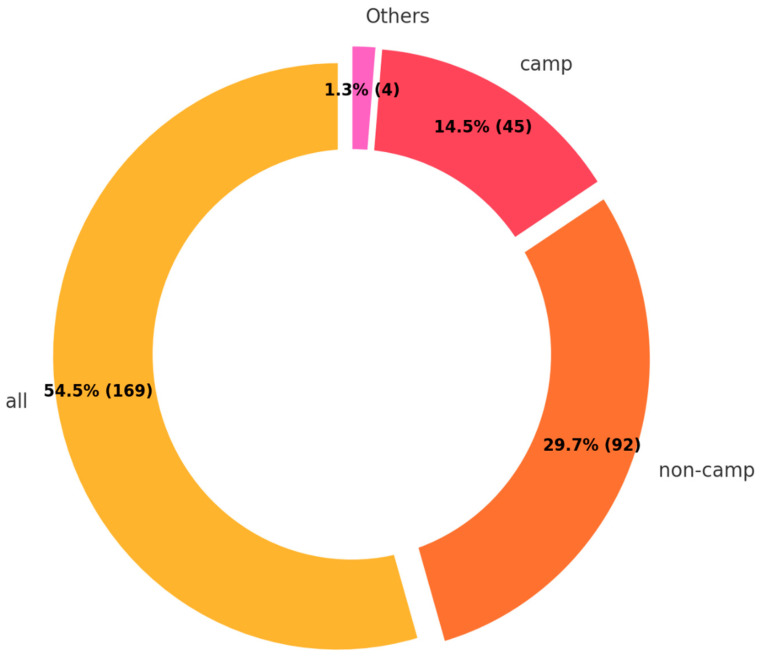
Living environments.

**Figure 12 ijerph-22-00063-f012:**
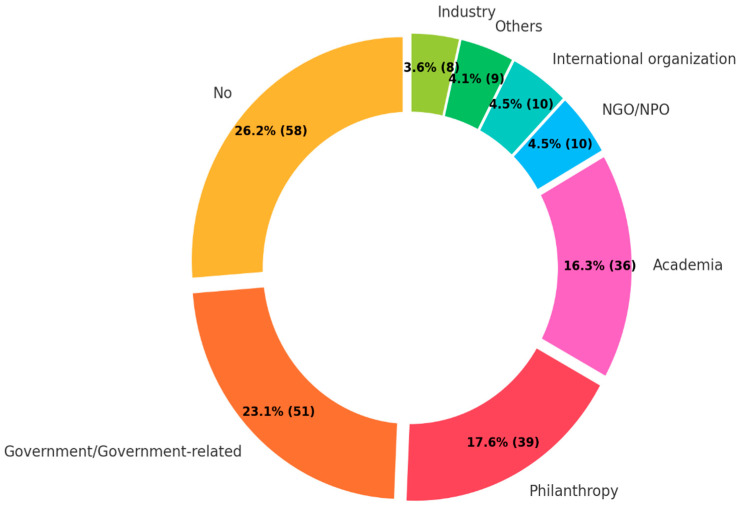
Funding source.

**Figure 13 ijerph-22-00063-f013:**
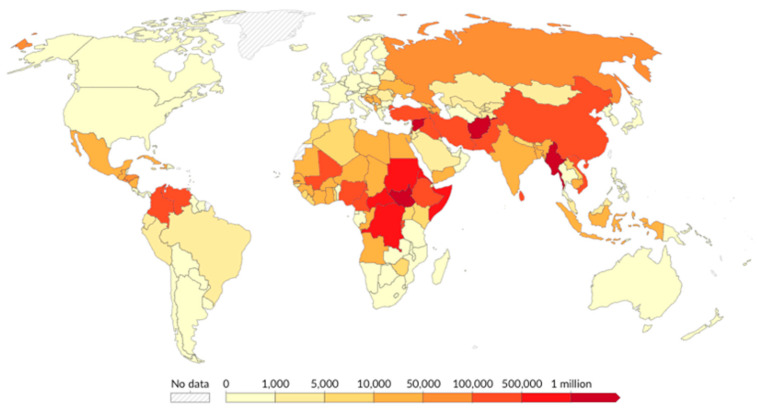
Refugee population by country or territory of origin, 2021.

**Figure 14 ijerph-22-00063-f014:**
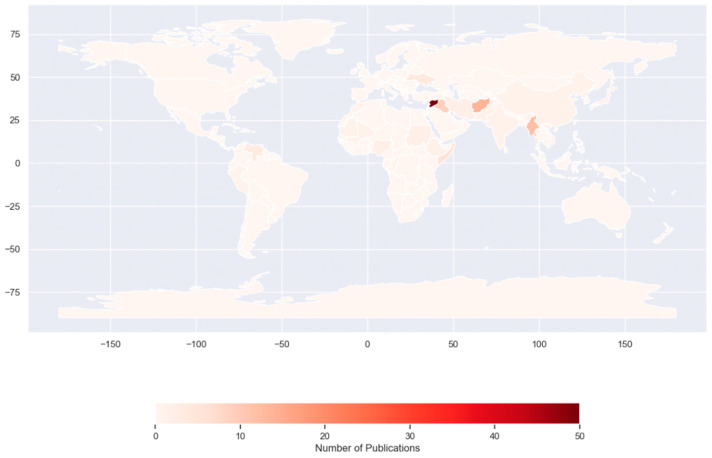
Number of publications by country of origin.

**Figure 15 ijerph-22-00063-f015:**
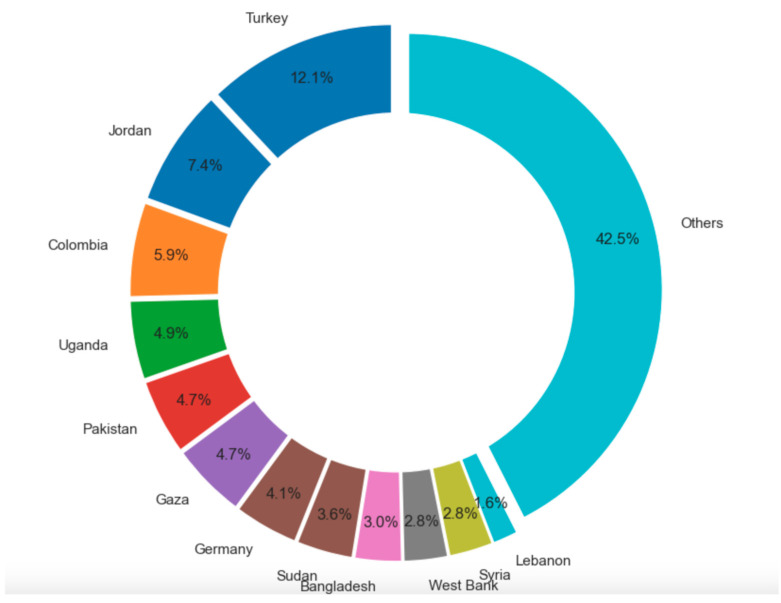
People displaced across borders by host country.

**Figure 16 ijerph-22-00063-f016:**
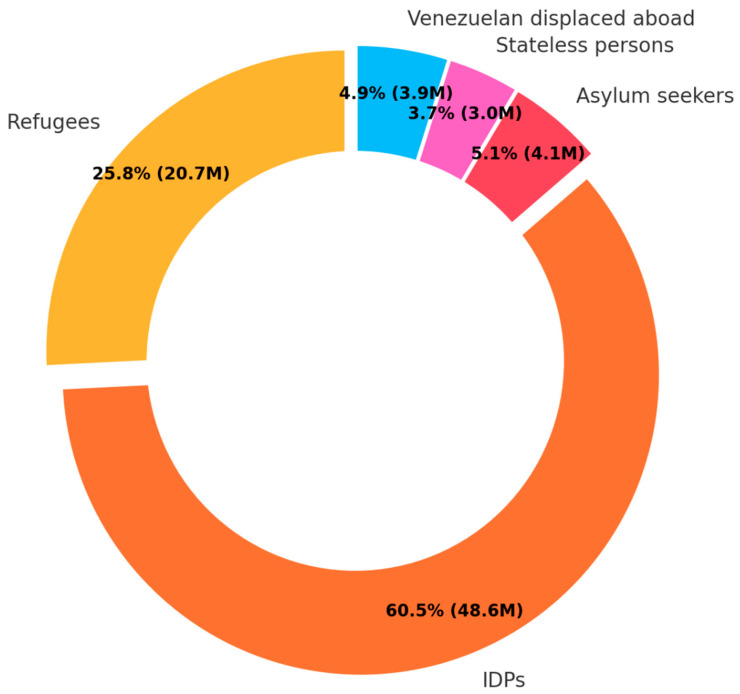
FDP by category in 2020.

**Table 1 ijerph-22-00063-t001:** Inclusion and exclusion criteria.

Parameter	Inclusion Criteria	Exclusion Criteria
Population	Refugees, IDP, Asylum seekers, Venezuelans displaced abroad, Stateless	Migrants not defined as FDP, Main target population is not FDP
Target diseases/ Research domain	1. Diseases: NCDs included in GBD study 2. Research domain: health assessment, health policy, healthcare system, patients’ experience	1. Diseases: Not included in NCDs in GBD study (e.g., communicable diseases, reproductive health, nutritional disorders, and injuries)
Study design	Classified into the level of scientific evidence I–VII, scoping review, literature review, cross-sectional study, ecological study	Not classified into the level of scientific evidence I–VII (e.g., news).
Publication types	Journal articles, thesis, and dissertations, organization reports	Books and book chapters, conference abstracts, study protocol
Language	English	Non-English
Study period	〜8 December 2022	9 December 2022〜

**Table 2 ijerph-22-00063-t002:** The level of scientific evidence.

Level of Evidence	Description
Level I	Evidence from a systematic review or meta-analysis of all relevant randomized controlled trials (RCTs)
Level II	Evidence from at least one well-designed RCT
Level III	Evidence from a single well-designed controlled trial without randomization
Level IV	Evidence from well-designed case–control or cohort studies
Level V	Evidence from a systematic review of descriptive and qualitative studies (meta-synthesis), scoping review, and other literature reviews
Level VI	Evidence from a descriptive or qualitative study, cross-sectional study, and ecological study
Level VII	Evidence from the opinion of authorities and/or reports of expert committees

## Data Availability

The original contributions presented in this study are included in the article/Appendix A. Further inquiries can be directed to the corresponding author.

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
