# Peer review of "Non-Communicable Diseases Among Forcibly Displaced People: A Systematic Mapping Review"

_ijerph, 2025, doi:10.3390/ijerph22010063_

Round 1

Reviewer 1 Report

Comments and Suggestions for Authors

Thank you for the opportunity to review this manuscript. After reading it, I have several major comments:

Background: The background offers useful context on NCDs, but the definition of Forcibly Displaced Populations (FDPs) is unclear. Please define who qualifies as an FDP (e.g., refugees, asylum seekers, IDPs) and explain how this relates to the study. Including specific examples or data to show gaps in understanding NCDs among FDPs, and explaining how the study addresses these gaps, would strengthen the research context.

Methodology: The methodology needs more transparency. Clearly explain study exclusions, the search strategy (databases, date range, filters), and the data extraction process. Following PRISMA-Scr guidelines would improve clarity. The search strategy seems incomplete, particularly regarding grey literature from humanitarian sources. Please provide examples of search terms or describe the keyword identification process.

Findings:

  1. The results section lacks citations. Please include citations from the studies referenced.
  2. It is unclear how the quality of the included studies was determined. How was the scientific evidence level assessed? This needs further explanation.
  3. In RQ 3, how were the percentages for each disease calculated? Was this an original calculation by the authors, or is it a report of results from other studies? Please clarify.
  4. The methods for creating Figures 14 and 15 are not explained. How were the percentages generated? 

Discussion: The discussion would benefit from comparing findings to broader NCD literature in general and vulnerable populations. Please reference key studies to position your findings within the existing research. I also disagree with the statement that ‘the predominant diseases studied are DM and hypertension.’ Figure 5 shows ‘Other NCDs’ as well, which should be mentioned and explained. Expanding on ‘Other NCDs’ would provide a more complete view of the NCD burden among FDPs. Finally, the mention of publication bias in the limitations section needs further clarification on how this was assessed. 

Comments on the Quality of English Language

Could be improved.

Author Response

Comment 1:

Background: The background offers useful context on NCDs, but the definition of Forcibly Displaced Populations (FDPs) is unclear. Please define who qualifies as an FDP (e.g., refugees, asylum seekers, IDPs) and explain how this relates to the study. Including specific examples or data to show gaps in understanding NCDs among FDPs, and explaining how the study addresses these gaps, would strengthen the research context.

Response 1:

Thank you for pointing this out. We agree with your comment and have added a clear definition of FDP with a reference. This change is highlighted in yellow on page 2, lines 63-65. The detailed types of FDP are mentioned in the Population parameter in the Inclusion and exclusion criteria (Table 1). 

Additionally, we agree to show gaps in understanding NCDs among FDP. This addition can be inserted into page 2, lines 79-85, with a reference highlighted in yellow.

Comment 2:

Methodology: The methodology needs more transparency. Clearly explain study exclusions, the search strategy (databases, date range, filters), and the data extraction process. Following PRISMA-Scr guidelines would improve clarity. The search strategy seems incomplete, particularly regarding grey literature from humanitarian sources. Please provide examples of search terms or describe the keyword identification process.

Response 2:

We agree that transparency in the methodology is important and have accordingly added a database search result to the Result section as Figure 1 on page 5.

Regarding the suggestion to follow RPISMA-Scr guidelines, we appreciate your recommendation. However, as this study is a systematic mapping review rather than systematic review, PRISMA-Scr guidelines are not directly applicable. Instead, we have adhered to a methodological framework specifically designed for systematic mapping reviews, which is better suited to the goals and scope our study.

Comment 3:

Findings:

  1. The results section lacks citations. Please include citations from the studies referenced.
  2. It is unclear how the quality of the included studies was determined. How was the scientific evidence level assessed? This needs further explanation.
  3. In RQ 3, how were the percentages for each disease calculated? Was this an original calculation by the authors, or is it a report of results from other studies? Please clarify.
  4. The methods for creating Figures 14 and 15 are not explained. How were the percentages generated? 

Response 3:

  1. Thank you for pointing this out. We have added the necessary citations to the Results section. These additions are highlited in yellow, inclucing citation 12 on page 7, line 206.
  2. We agree that a detailed explanation of the scientific evidence level is important. To address this, we have provided it as Table 2 on page4, which was highlited in yellow.
  3. Regarding the calculation of percentages for each disease in RQ3, there was a difference between the sentences and Figure 6. Figure 6 presents the percentage based on all diseases mentioned in the retrieved literature (n=390). Meanwhile, the percentages in the sentences were calculated excluding literature which described a comprehensive overview of NCDs without sepcifying individual diseases (n=327). We calculated this result using data from the literature retrieved for this mapping review.
  4. Figure 14 and 15 (renumbered as Figure 15 and 16 in revision 2) were developed using from Citation 14 and Citation 8, respectively.

Comment4:

Discussion: The discussion would benefit from comparing findings to broader NCD literature in general and vulnerable populations. Please reference key studies to position your findings within the existing research. I also disagree with the statement that ‘the predominant diseases studied are DM and hypertension.’ Figure 5 shows ‘Other NCDs’ as well, which should be mentioned and explained. Expanding on ‘Other NCDs’ would provide a more complete view of the NCD burden among FDPs. Finally, the mention of publication bias in the limitations section needs further clarification on how this was assessed. 

Response 4:

Thank you for your valuable insights and suggestions. We agree that comparing our findings to the broader NCD literaterue in general and vulnerable population is an important aspect.  To address this, we have added a comparison with broader NCD literature in general under “Gap3. Target diseases” on page 17, lines 381-389, highlighted in yellow. Additionally,  we included comments on vulnerable populations in the Introduction section on page 2, line 79-85, also highlighted in yellow.

Regarding “Other NCDs”, we have listed some of them on page 8, lines 234-236, to provide clearer understanding. All diseases included in the review are recorded in the S1 Appendix for further references.

As for publication bias, we have provided further clarification on why it is a concern and how it may have impacted our findings on page 19, lines 429-434, highlighted in yellow.

Reviewer 2 Report

Comments and Suggestions for Authors

Line 13 - The circumstances of the region of origin of FDP are always sad and unfortunate. That means that whatever is said about FDP would generally apply to the remaining population (who don't become displaced).  While it is not the subject of your study, perhaps there should be at least some acknowledgment of their parlous conditions and health.

Line 16 - In addition, shouldn't you use medical journals such as The Lancet and BMJ?  They would likely be a source of valuable data that may have been missed by those sources you used.

Line 26 - "...current knowledge areas of focus and gaps".  (I think this sounds better.)

Line 129 - You mention six aspects but only list five aspects immediately following.  I think you have omitted "Study Period"

Lines 166-169 - Please check the wording as you don't appear to have listed nine key variables.

Line 193 - (Based on a hierarchy of I - VII in descending order.)

Line 253 - This needs clarification as to the meaning.

Line 295 - When you say "government" do you mean the government of the country in which the research authors are based? If that is the case, what about when co-authors in a research project are based in two or more countries?

Line 410 - There appears to be another category of limitation that you may need to consider - and that is FDP who have been resettled in other countries where their wellbeing is being met by the health system of the jurisdiction of their new place of residence.

Author Response

Comments 1: Line 13 - The circumstances of the region of origin of FDP are always sad and unfortunate. That means that whatever is said about FDP would generally apply to the remaining population (who don't become displaced).  While it is not the subject of your study, perhaps there should be at least some acknowledgement of their parlous conditions and health.

Response 1: Thank you for your comments on the issue. We agree with your observation and have made the necessary acknowledgment . We added the acknowledgement on page 2, line 68, highlighted in yellow.

Comments 2: Line 16 - In addition, shouldn't you use medical journals such as The Lancet and BMJ?  They would likely be a source of valuable data that may have been missed by those sources you used.

Response 2: Thank you for your suggestions. Medical journals, including The Lancet and BMJ,  are indeed valuable sources for the mapping review. As the five databases we used cover these journals, their data were incorporated into the review.

Comments 3: Line 26 - "...current knowledge areas of focus and gaps".  (I think this sounds better.)

Response 3: Thank you for your suggestion. We have revised the sentence as recommended, and the change is highlighed in yellow.

Comments 4: Line 129 - You mention six aspects but only list five aspects immediately following.  I think you have omitted "Study Period"

Response 4: Thank you for pointing this out, and we apologize for the confusion. The six aspects include population, target diseases, research domain, study design, publication types, and languages , as described in table 1. The “study period” aspect refers only to our research timeframe. To avoid further confusion, we have clarified this by revising the sentence to include “target disease” explicitly, which is highlighed in yellow on page 3, line 129.

Comments 5: Lines 166-169 - Please check the wording as you don't appear to have listed nine key variables.

Response 5: We again apologize for the confusion. We guess the misunderstanding may stem from counting “countries of origin and host” as a single variable. To clarify, we have revised the sentence to list “countries of origin” and “countries of host” as separate variables, and highlighted in yellow.

Comments 6: Line 193 - (Based on a hierarchy of I - VII in descending order.)

Response 6: Thank you for emphasizing the need to clarify the hierarchyb of evidence levels. We have added a table (Table 2) on page 4, defining each scientific evidence level.

Comments 7: Line 253 - This needs clarification as to the meaning.

Response 7: Thank you for your comment. While the topic may suggest some usefuln implications, the purpose of the Result section is to present the data objectively without interpretation. We will explore this topic further in future research.

Comments 8: Line 295 - When you say "government" do you mean the government of the country in which the research authors are based? If that is the case, what about when co-authors in a research project are based in two or more countries?

Response 8: Thank you for your question. In this context, “government” refers to funding provided by a government or government-related funding agency, regardless of the nationalities of the authors involved in the research.

Comments 9: Line 410 - There appears to be another category of limitation that you may need to consider - and that is FDP who have been resettled in other countries where their wellbeing is being met by the health system of the jurisdiction of their new place of residence.

Response 9: Thank you for highlighting this important concern. It is true that FDP face challenges in resettling into host communities. We incorporated your suggestion on page 19, lines 410-412, highlighted in yellow.
